# Online suicidal thoughts and/or behaviours talk: A scoping review protocol

**Andrea Lamont-Mills** [ID]*, **Luke T. Bayliss** [ID], **Steven A. Christensen**

School of Psychology and Wellbeing, University of Southern Queensland, Ipswich, Queensland, Australia

* andrea.lamont-mills@usq.edu.au

## Abstract

The anonymity that the internet and social media affords users means that suicidal thoughts and/or behaviours can be talked about with a sense of freedom and disinhibition that is often not possible in face-to-face contexts. Better understanding online suicidal thoughts and/or behaviour talk is critical as more people turn to online spaces for support. Without this the potentiality of such spaces as sites for suicide prevention and intervention is likely to remain unrealised. Currently there are no scoping or systematic review syntheses focusing on internet and/or on social media suicidal thoughts and/or behaviour talk. This lack of synthesis is problematic as it makes it more difficult for online suicide prevention and intervention practices, policies, and our understanding of suicide to advance in a coherent and evidence-based manner. A scoping review protocol following Arksey and O'Malley's six-step modified framework has been developed to address this synthesis gap. It aims to systematically map the empirical literature that has investigated online suicidal thoughts and/or behaviours talk. It is anticipated that review outcomes could inform the training of health practitioners and peer/professional online moderators in how to best talk with people experiencing suicidal thoughts and/or behaviours. Outcomes could also form an evidence-base for developing policies and practices that focus on online places as safe spaces to talk about suicidal thoughts and/or behaviours. Developers of safe language guidelines could also use the outcomes to audit how well current guidelines reflect empirical evidence. Outcomes could enable researchers to design future online suicidal thoughts and/behaviours talk studies that extend our understandings of suicide leading to potential refinements of contemporary suicide theories/models.

## Introduction

Despite suicide prevention and suicidology research efforts, suicide remains a global public health issue. Access to formal health care and support is critical for preventing deaths by suicide, however many people contemplating a suicide attempt do not seek formal health care and support [1]. This can be because of access difficulties [2], cost, [3], feelings of shame [4], concerns with being stigmatised [5], or fears of being hospitalised [6]. Many turn instead to the internet and/or social media for help.

**Data Availability Statement:** No datasets were generated or analysed during the current study. This is a study protocol.

**Funding:** The author(s) received no specific funding for this work.

**Competing interests:** The authors have declared that no competing interests exist.

There is conjecture about what is meant by the terms internet and social media. Broadly, internet relates to internet based applications (e.g., chat rooms, online forums, websites) that allow for content creation and sharing with the potentiality of user engagement with the posted content and by default other users [7]. Social media refers to mobile applications (e.g., Facebook, Twitter) that also enables users to create content and share this with others who can engage with this user generated content and the user themselves [7].

Whilst the above is suggestive of the interactive nature of both online spaces, the internet and social media are often portrayed as being primarily information sources or repositories [8]. That is, they are online spaces where users go for information and support. Here users are conceptualised as passive information/support seekers [9], which is sometimes reinforced by online suicidology research. This is because individual user experiences or perceptions are privileged, with research typically focusing on the impact that using the internet and social media has on the individual user's suicidal thoughts and/or behaviours (STBs). Yet these online spaces are more than simply information sources or repositories, they are where users go to engage and interact (i.e., talk) with others and to share their experiences of STBs [8]. In suicidology research the engagement and interaction of users in these online spaces is often overlooked with preference being given to user impact of going online for suicide-related reasons. This is not dismissing the research that has focused on the supportive nature of online spaces for those experiencing STBs (see [10–12]). Rather the point is such research has not necessarily been concerned with actual online STBs talk or how users engage and interact with others in these online spaces. Instead, most of this research focuses on individual user benefits or negative impacts.

Suicide and STBs are contested terms with little previous consensus on what they encompass and mean [13]. Drawing upon the recent work of De Leo and colleagues [13], suicide is a fatal act that is carried out with knowledge of this fatality. Suicidal ideation encompasses thoughts of killing oneself where there may or may not be an intention to take one's own life. It is the absence of behaviour that distinguishes suicidal thoughts from suicide and suicidal behaviour, noting that suicide in and of itself is a behaviour. Taking a broad perspective on suicidal behaviour, this refers to having made plans or preparing to take one's own life that includes the how and when of this but also extends to having made a suicide attempt [13]. It is perhaps with self-harm that most disagreement arises, centring around the issue of intention. For our purposes, self-harm is when an individual engages in activities that are harmful to themself but there is an absence of an intention to die, thus the activities are non-fatal [13].

Moving to online talk, we define online talk as the digital text-based language people use when communicating to and with others, either in real-time or asynchronously [14]. This is via mediums that include but are not limited to online forums, direct messaging apps or blogs. Online talk therefore includes the posts and reply comments people make on such forums, the messages that they send on direct messaging apps, and what they write on blogs and in blog comments. In this sense online STBs talk encompasses written text about STBs be this one's own or others. Taking up an online STBs talk focus does not mean that people cannot and do not communicate STBs visually (e.g., via images, videos, memes [15, 16]). Communicating via text and visual methods draws upon different understandings and competencies and are often used for different communication purposes [17]. They are two related but distinctive ways of communicating and as such visual communication of STBs requires its own separate consideration. This is particularly so as many online spaces restrict STBs images, videos, and memes being shared [18] but are less restrictive regarding online STBs talk.

Talking about STBs in online spaces brings with it a sense of freedom and anonymity that is often not possible in formal health care settings [19]. The social restrictions and inhibitions that are often present in face-to-face therapy are less constrictive in online spaces. This is called

the online disinhibition effect where individuals disclose more online than they usually would in face-to-face environments [20]. Talking about STBs online is, therefore, likely to be different to how STBs are talked about in offline spaces. Individuals who are in a heightened state of suicidal desire often withhold or are reluctant to share information in face-to-face or real-time settings and feel they are able to discuss matters that concern them more freely in online spaces.

Validation of experiences and a sense of belonging, connection, and support can be gained through engaging with other users in online spaces [21]. Whilst what engagement means is unclear, Thoër [8] argues that this enhanced sense of belonging and connection comes from users talking with each other about their STBs. A sense of belonging is theorised as a protective factor in contemporary theories of suicide [22–24]. Focusing on online STBs talk is a way researchers can examine what is occurring in these spaces, how users support and influence each other, how users keep each other safe, and how any effect of participation in these spaces is achieved.

This speaks to the centrality of online engagement and interaction and the importance of considering not only what users say about STBs but also how are they talking about and discussing STBs with others. This centrality is already evident in suicide prevention and intervention approaches. Talking with a person who is in a heightened state of suicidal desire in order to keep them safe is at the heart of most suicide prevention programs such as gatekeeper training [25] and mental health first aid [26]. Moreover, the importance of talk is reflected in safe language guidelines. These have been developed to ensure that the stigma and shame surrounding suicide is not being perpetuated through language (see #chatsafe guidelines [27]; Mindframe language guidelines [28]; CAMH words matter: Suicide language guide [29]). Further, asking and talking about suicide is noted as being difficult with this being reflected in the number of suicide prevention training programs specifically designed to assist 'ordinary' people to talk with someone who is in a state of heighted suicidal desire (e.g., safetalk [30], calm conversations [31] and zero suicide alliance suicide awareness training [32]).

Gaining or giving lifesaving support and assistance is not easily accomplished without interacting with the individual who is experiencing a heightened state of suicidal desire. The same occurs in online spaces, they are places where individuals who are experiencing STBs go to connect with others and talk about their distress. In doing so they can be moved away from the 'edge of suicide' [33] and toward a sense of belonging and optimism [34]. Therefore, identifying what research has been conducted regarding online STBs talk, and what this research has found is needed. Without this, the potentiality of such research to identify better ways to provide support to those experiencing a heightened sense of suicidal desire, to inform the development of internet and social media suicide prevention and intervention programs, and to advance our knowledge of suicide will go unrealised.

There is concern about people using the internet and social media for suicide-related purposes. There is evidence for both the potential benefits (e.g., reduction in suicidal thoughts after using online forums [35]) and negative consequences (e.g., increased feelings of hopelessness [36]) of such use. Further, there is the potential that reading and/or commenting on others online STBs talk can be triggering as there can be contagion effects of viewing and engaging with others online and with online STBs content [37]. These benefits and negative impacts aside, the issue with usage-focused research is that there is ambiguity around what use actually means. Different studies conceptualise use differently, such as the number of times online forums or social media platforms are visited [37] or has the internet been used for suicide-related purposes [38], with most studies conceptualising use as the amount of time/number of days spent online [39, 40]. However, use can also be conceptualised as online STBs talk

which includes what people say, how they say this, and how others engage and interact with this online STBs talk [8].

There have been number of scoping and systematic reviews focusing on STBs in terms of internet and communication technology or internet and social media use. The technology reviews have focused on evidence for the effectiveness of apps as alternatives for managing self-harm and suicidal thoughts [41] or have sought to identify best technology practices and programs for suicide prevention [42]. Internet reviews have either focused on online activity use and its relationship to self-harm and suicidal behaviour in young people [21], online use and the exacerbation of self-harm and suicide risk in young adults [36], have broadly mapped the development of internet suicide research [43], or have focused on online content in terms of user experience with content, knowledge of content, or the actual content itself [44]. The benefit of young adults social media use in relation to self-harm or suicidal ideation [45], suicide risk exacerbation and social media use [46], Instagram self-harm and suicide content [47], and how social media is used for suicide prevention [48] have been the focus of the social media reviews. Young people and young adults, suicide prevention, online content, and use benefits/risks appear to be common review focal points. Besides the Robinson et al [48] social media review which sought to identify evidence for social media use and suicide prevention, all other reviews overlook and therefore have not included research that has examined online STBs talk.

As part of the Robinson et al [48] review, research that examined internet and social media user language was included as part of the evidence addressing the review's aim. Of the four studies identified, three investigated how users expressed themselves as suicidal, with single forum posts or online letters being the data [49–51]. One study looked at the interaction between users and how users, through interacting with each other, established themselves as suicidal [33]. Whilst this talk focus is encouraging, the suicide prevention focus of the review means that findings are related to social media sites that specifically aim to provide support to people experiencing STBs. Because of the clear suicide prevention focus, research that sought to examine talk on other types of online spaces (i.e., pro-choice/pro-death, general health sites) were not included in the review. Overall, the review was more focused on identifying research regarding the benefits of social media use rather than online STBs talk research. Further, only studies up to 2014 were included in the review and as more talk-based research has been conducted since then, this raises questions about the currency of these findings. This is particularly so given the increased use of the Internet and social media for health related purposes over the last few years [52], including older people who have increased their use of the internet and social media in general [53, 54].

Currently there are no scoping or systematic review syntheses of research studies focusing on online STBs talk. A lack of a comprehensive overview of evidence means that the potential of internet and social media platforms to inform suicide prevention and intervention approaches remains under-realised. Further, internet and social media technology has developed and changed rapidly in the past 5 to 10 years with more mental health organisations offering online suicide support. This with the more recent development of peer-led online suicide prevention programs [55], raises questions about the utility of prior systematic and scoping reviews as platforms for understanding how to best engage people who are experiencing STBs and turning to online places for support and help.

A scoping review is proposed to address the aforementioned review synthesis absence. Scoping reviews are well suited for collating evidence of emerging areas of research [56] and is suitable for this particular research context. The aim of the proposed scoping review is to systematically map the empirical literature that has investigated online STBs talk. By collating and mapping current evidence, knowledge gaps and study limitations can be identified that can be

used to advance future practice, policy, and research. Therefore, the aim of this paper is to present an a priori scoping review protocol for critical assessment in terms of bias identification, rigour, and transparency of the proposed scoping review methodology. Providing validation for the scoping review process will increase the potential of the scoping review to make a meaningful contribution to advancing practice, policy, and research relating to online STBs talk.

## Materials and methods

Arksey and O'Malley's [56] six-step scoping review framework and recommended modifications [57, 58] have been used to develop this protocol. The six steps and modifications have been outlined below. The protocol is also registered with Open Science Framework [59].

### Step 1 –Identifying the research question

The research question, purposes, and objectives have been developed using the Population/ Participant–Concept–Context [60] mnemonic (Table 1). This adds clarity to, and alignment between, the research question and protocol purposes and objectives [57].

The overall research question is deliberately broad and seeks to identify what online STB talk research has been conducted. There are three purposes associated with this review, with each purpose related to a specific objective. The first is to map the range, nature, and extent of research conducted on online STBs talk. Doing this will allow the field to advance in a systematic, evidence-based manner increasing the potential of review findings to inform suicide prevention and intervention practices, policies, and research.

**Table 1. Population/Participant–Concept–Context parameters with inclusion and exclusion criteria.**

| Mnemonic | Mnemonic parameters | Inclusion criteria | Exclusion criteria |
|---|---|---|---|
| **Population/ participants** | Individuals of all ages, genders, sexual orientations, or nationalities with current or past experiences of STBs. Behaviours are suicidal plans, preparations, and/or attempts [13]. Thoughts relate to ideas and ruminations of taking one's own life where intention may or may not be present [13]. | Research studies that contain a measure of, or provide evidence that, research participants have current or past experiences of STBs. Evidence may be targeted recruitment, self-identification, or other identification (e.g., psychologist). Measures may include but are not limited to the Suicide Behaviors Questionnaire–Revised [61], Suicidal Ideation Attributes Scale [62], or the Beck Scale for Suicide Ideation [63] | Research studies that contain a measure of, or provide evidence that, research participants only have current or past experiences of self-harm behaviours. Self-harm is a distinct and separate concept from STBs. The distinction is that when engaging in harm to self, there is no intention to take one's own life [13, 64]. Thus studies that have measures of self-harm that have items relating to STBs (e.g., Self-Harm Inventory [65]) would be included but studies that use measures that do not (e.g., Self-Harm Behavior survey [66]) would not be included. It is the absence of a measure of STB that would exclude a study. |
| **Concept** | Online STBs talk. | Research studies that include a result relating to online STBs talk. Data that would have produced this result would include but are not limited to online posts/Tweets/comments, online letters, or interactions between users themselves or between users and moderators. This would also include results where users have commented on online visual representations of STBs (e.g., studies that have analysed comments on You Tube videos). | Research studies that include only results relating to users talking about STBs with health professionals. Studies that have only analysed online visual images such as photos, videos, memes. That is, the analysis is a visual rather than textual analysis. Opinion pieces, Editorials, book reviews and other non-research articles will be excluded. |
| **Context** | Study settings are internet and social media spaces. These are spaces that allow for content creation and sharing with the potentiality of user engagement with the posted content and by default other users [7] | Research studies that include a result relating to online STB talk. Places where this talk may have originated include but are not limited to online discussion forums, online message boards, online discussion bulletins, Twitter, Instagram, Blogs, myspace, Snapchat, Weibo, and Facebook. Other spaces are likely to be identified and added to as an outcome of step 2. | Research studies that focus only on telehealth or offline spaces as places where STBs are discussed. |

The second purpose is to disseminate what is known about online STBs talk. This may enhance the training of health practitioners and peer/professional online moderators in suicide prevention strategies. Specifically, this is in terms of how to best talk with people experiencing STBs. Dissemination extends to improving suicide prevention and intervention policies and practices to enable online spaces to be safe places to talk about STBs. Developers of safe language guidelines may also use findings to audit how well current guidelines reflect empirical evidence. The final purpose is to identify gaps in the existing knowledge base that may be used to guide future online STBs talk research in a systematic manner. Findings may also be used to refine understandings of contemporary suicidal theories/models.

## Step 2 –Identifying relevant studies

To identify relevant studies, a three-stage approach will be undertaken with the first stage having been conducted as part of the protocol development. Open Science Framework, Prospero, Cochrane, and JBI Evidence Synthesis were searched by reviewer 1 (the first author) on the 18[th] of March 2022 to identify any existing systematic or scoping reviews that reflect the proposed review's research question. No scoping or other type of systematic review was identified. The second stage has already been conducted with reviewer 1 undertaking a preliminary search of the EbscoHost Academic Search Ultimate database to test the search strings on the 18[th] of March 2022. Subject terms and keywords from the first 30 returned articles have been scanned to identify if there were any key search terms that were not included in the search string strategy or were not relevant to the search string strategy. No new search terms or potential changes to the search string were identified. Based on reviewer feedback Snapchat and Weibo were added to the second search string. The final stage will be for reviewer 1 and 2 (the second author) to independently search all the databases, key journals, and reference lists of included articles.

The search strategy (see S1 File) and electronic database selection has been developed in consultation with a research librarian who has expertise in systematic review study database and search string identification and development. No language limitation will be imposed during database searching. Should studies in a language other than English be identified, a translated version of the title and abstract will be sought from the corresponding author of the study, if such a translation has not been identified via a search of the selected databases and journal website where the study is published. If a translated version is not available, the study will be included in a supplementary table and noted as a potential limitation of the review's findings. Translation apps or software will not be used to translate study titles or abstracts. Research suggests that they are useful for simple everyday phrase translation with error rate increasing with sentence complexity [67]. Such complexity is likely to be found in research article titles and abstracts. A date limitation of 1989 onward will be imposed during this search stage that reflects when the internet became publicly available.

A quality assessment limitation will also be implemented during the search stage. Although quality or critical assessment of studies is not a scoping review requirement [56, 57], some published scoping review protocols [68, 69] and reviews [70–73] include such an assessment. Further, as one of the purposes of the proposed review is to identify gaps in the literature, quality assessment has been recommended in such situations [74]. Therefore, a two-phase quality assessment process has been adopted. The first will be to impose a search limitation of only peer reviewed journals during the search phase, where possible [75]. This means that only peer reviewed research studies will be included in this review. The second is outlined at step 4.

A tension for any scoping review occurs between the feasibility of doing the review and its breadth. The need for a comprehensive mapping of the field needs to be balanced against

practicalities so that it can be conducted in such a way that it meets the review's purpose and objectives in a timely manner [57]. The databases have been chosen to reflect this in that they that are health orientated as well as sociological/discursive/language focused databases. This is because the substantive area is both health (STBs) and sociological/discursive (talk, language, conversations, or interactions). The databases have also been chosen with a view to identifying relevant studies whilst minimising study duplication.

Database searching will be undertaken independently by reviewer 1 and reviewer 2 on the same day. Databases will be searched one at a time in the following order Medline, CINAHL with Full Text, Academic Search Ultimate, APA PsycInfo, Communication Source, Health Source: Nursing/Academic Edition, Humanities Source Ultimate, MLA Directory of Periodicals, Psychology and Behavioural Sciences Collection, Sociology Source Ultimate, Scopus, and SAGE Journals: Social Sciences and Humanities to minimise duplicates [76]. Searches will be conducted by title and then by abstract, where possible. For each database search, search string 1 will be run first, following by search string 2, then search string 3, and then search string 4. The outcomes of the three separate searches will be combined using the Boolean operator AND to become the final search.

Both reviewers will note the outcome of each database search in a separate Microsoft Excel file that will act as a review audit trail (see S2 File). After each database title and abstract search, reviewer 1 and reviewer 2, will compare results. Should discrepancy arise, the reviewers will discuss this in order to identify the source of the discrepancy. Discrepancy identification will be noted on the Microsoft Excel review audit trail files with a corrected search then run. If discrepancy identification fails, a third reviewer (the third author) will overview reviewer 1 and 2's search processes to identify the discrepancy. This outcome will also be noted on the file. This general comparison-discrepancy-consensus process will be used throughout the review, unless otherwise stated.

Besides electronic databases, reference lists of studies identified at the end of stage 3 will be independently searched by reviewer 1 and 2 to ensure that all potentially relevant studies have been considered for this review. Identified studies will be considered for inclusion using the same inclusion-exclusion criteria (see Table 1) as studies identified via database searching. The reviewers will compare results after searching all reference lists with outcomes being noted on the Microsoft Excel review audit trail files.

A separate independent search of key suicide, discursive, and cyberpsychology research journals will also be undertaken by reviewer 1 and 2. This is required because electronic databases have a British and American journal bias [56]. Key suicide journals to be searched are Suicide and Life-Threatening Behaviors, Archives of Suicide Research, Death Studies, and Crisis: The Journal of Crisis Intervention and Suicide Prevention. Key discursive/language research journals are Sociology of Health and Illness, Qualitative Health Research, Discourse Studies, Journal of Pragmatics, and Discourse and Society. The key cyberpsychology journals are Cyberpsychology, Behavior and Social Networking and Computers in Human Behavior. Other key journals may be added depending on search results. The same result comparison process will occur after all journals have been searched with results being noted on the Microsoft Excel review audit trail files.

Given the inclusion of a two-phase quality assessment process, a search of the grey literature will not be undertaken. This is not only related to quality assessment but also due to the difficulty in identifying what grey literature to include and how to do this in a systematic and reproducible manner. For example, mental health or suicide prevention organisations may have commissioned research to investigate online talk. Typically, the websites of such organisations would be searched for such studies with these organisations being identified using an internet search engine. The issue with using an internet search engine is that the same search

conducted by two different reviewers will return different results [77]. Indeed, the same search conducted by the same reviewer on different occasions will also return different results. The algorithms that underpin such a search, returns results that are bespoke to the particular reviewer's preferences which means the algorithms are constantly changing and adapting [77]. Any results from such a search are not systematic nor reproducible and thus not appropriate to be considered for a scoping review.

Whilst including a search of the grey literature has been a notable feature of scoping reviews [56, 58], there are no clear guidelines as to what grey literature must be included in a scoping review. Further, there are no best practice recommendations as to how to search for and thus identify all relevant grey literature sources [78]. Currently it is recommended that overlapping search strategies be adopted in order to identify all relevant pieces of grey literature, however this brings with it the probability of also identifying irrelevant literature [78]. Thus, it is not only replicability that becomes an issue but as Peters and colleagues [58] posit the scope and breadth of the review needs to be balanced alongside the feasibility of completing the review with the resources available. We recognise that the exclusion of grey literature will impose limitations on review findings, and this will be noted in limitation discussions. As per Peters et al. [58, 79] recommendation when deciding to not include particular searches, we have justified, detailed, and made transparent our decision to not include grey literature and recognise this as a limitation.

The results of the above database, reference list, and key journals searches will be managed using EndNote X9 [80]. Each reviewer will manage their own search results thus two independent EndNote libraries will be developed. Each library will have a separate EndNote group folder for each database search and for each search type (journal search, reference list search).

## Step 3 –Study selection

Reviewers 1 and 2 will independently engage in the below study selection stages using the inclusion and exclusion criteria presented in Table 1. Team members have met and discussed the appropriateness of the databases and search strings as well as the inclusion and exclusion criteria [57]. Team meetings will be held regularly throughout step 3 to ensure that inclusion and exclusion criteria remain relevant as this allows for issues and challenges to be identified early and acted upon as required. The first meeting will be before study selection begins with the second occurring after the first half of the identified studies have been reviewed. Any changes to the inclusion and exclusion criteria will be through discussion and consensus with changes noted in the Microsoft Excel search review audit trail files. Given the iterative nature of scoping reviews [56], any changes may require reviewer 1 and 2 to restart the study selection process using any new updated inclusion and exclusion criteria. The final meeting will occur at the end of the study selection process to review the study selection process and to identify any additional changes that may be required to step 2 or 3. Any changes are likely to require reviewer 1 and 2 to re-visit steps 2 and 3.

The first stage in study selection will be for reviewer 1 and 2 to independently identify duplicate results using the Find Duplicates function in EndNote. The reviewers will do this following the systematic 13 step process outlined by Falconer [76]. The reviewers will meet after this to confirm remaining study numbers. Any discrepancies will be managed using the aforementioned discrepancy process. Once duplicates have been removed, reviewers 1 and 2 will independently apply the inclusion and exclusion criteria to both title and abstract at the same time. Reviewer 1's experience with previous reviews [69–71] indicates that a two-phase study title and then study abstract screening has the potential to miss studies that should be included in the final review. Given this, combining title and abstract screening is more feasible and less likely to exclude studies that should be included.

In stage 2, reviewers 1 and 2 will compare results after screening all studies where the first author has a last name starting with A, they then will compare after screening studies where the first author has a last name starting with B, with this continuing through to Z. This way any errors, issues and challenges are identified as they occur and can be acted upon with immediacy. The same consensus and discrepancy process that occurred in previous steps will continue during this screening process. One difference is for any studies where both reviewers are unsure whether to exclude or include. When such instances occur, the study will be included for full text screening.

Once the title and abstract stage screening has finished, the full text of remaining studies will be read independently by each reviewer. Using the inclusion and exclusion criteria, each full text study will be assessed for final review inclusion. The reviewers will meet after reviewing all full-text articles to compare results. After which, studies to be included for final review will have been identified.

## Step 4 –Data extraction

A Microsoft Excel data extract template based on the JBI data charting template has already been developed in consultation with all team members (S3 File). It contains both general (e.g., authors, publication year, location of study) and study specific (e.g., type of social media platform or online space, data, moderated/unmoderated online space, suicidality–thoughts and/or behaviours, limitations) information. Extracted information will align with the research question and purpose and objectives of the review. The template will be trialled independently by reviewer 1 and 2 with the first 5 articles for usability [57]. The research team will discuss the outcomes of this, with any changes being agreed to by all team members. As review and refinement of the data extraction template is likely to take place during this step [57]. There is the possibility that study information will be re-extracted as the extraction form evolves.

Reviewer 1 will engage in extracting data from the remaining studies with reviewer 2 undertaking an audit of a random selection of 20% of the remaining extractions to check for errors, omissions, and biases. The 20% figure has been used in a number of reviews as an appropriate study audit percentage [70, 71, 75]. This audit also offers an opportunity for further refinement of the extraction form. Outcomes of the audit will be charted on the Microsoft Excel review audit trail file with the same previously mentioned discussion and adjudication process being adopted. Reviewer 1 will contact corresponding author/s if relevant information has not been reported in the article, prior to extraction and audit taking place.

Stage 4 is where the second quality assessment will occur. Reviewer 1 will undertake a quality assessment of each included study during the data extraction process. Studies that are qualitative will be assessed using a modified JBI Appraisal Checklist for Qualitative Research [81]. Quantitative studies will be assessed using a modification of the appropriate JBI appraisal checklist (e.g., cross sectional studies will be assessed using the JBI Appraisal Checklist Analytical Cross-Sectional Studies [82], case reports the JBI Appraisal Checklist Analytical Case Report [83]). The JBI checklists have been chosen as they have been developed using an evidence-based approach, with the JBI Scientific Committee giving their approval for use. The checklists have been modified by reviewer 1 with changes being agreed to by the research team. These changes are based on Macrynikola et al [46] recommendations for selection, measurement and confounds. In terms of selection: did the study include clear sample descriptions/definitions such as age, gender orientation, nationality, sexual orientation, geographical location; was a participation rate included (if relevant) and was the recruitment criteria applied appropriately and uniformly; and was a justification of the sample size included. Measurement focused on the quality of measurement used in each study with a rating of high to low given

(see S4 File) with confounds focusing on whether basic and potential confounds that have been identified in the literature were taken into consideration and discussed.

This review will adopt a 80%, 50%-80% and below 50% rating process that previous reviews have used [68, 70, 71]. For each item on the checklist, 1 point is allocated to a 'yes' response and 0 for a no. When more than 80% of the checklist criteria has been met, the study is assessed as being of good methodological quality, moderate quality studies fall between the 50% to 80% range, and below 50% is indicative of poor methodological quality. Quality assessment outcome will be noted on the relevant data extraction form. Once this is completed the results of all the extraction and quality assessment processes will be entered into one Microsoft Excel file so that the key study aspects can be more easily identified, compared, and summarised.

## Step 5 –Collating, synthesising, and reporting the results

Stage 5 is where research that has been identified for final inclusion will be collated, synthesised, and reported in such a way that it answers the research question and meets the review's purposes and objectives. To assist with this, article selection stages and processes will be visually represented using the PRISMA flow chart (S1 Fig) with the PRISMA-ScR checklist (S5 File) being used to guide the overall conduct and reporting of the scoping review's process and findings. Based on previous pilot searches, both quantitative and qualitative studies are likely to be included in the results of this review. A textual narrative synthesis is the most likely methodological approach that will be adopted as this is well suited for synthesising both qualitative and quantitative research evidence as it allows for a standardise reporting of each type of study [84]. This is turn makes it easier for differences and similarities between study contexts and characteristics to be identified [85].

Synthesis will be undertaken by reviewer 1 using a step-wise approach [84]. The first step is placing studies into broad sub-groups. Sub-groups are likely to be internet, social media, non-attempting suicidal ideators, and attempting ideators. Studies can and are likely to be grouped in more than one sub-group with the possibility of other sub-groups emerging or current sub-groups being refined (e.g., internet–moderated online forums, internet–unmoderated online forums, social media–Twitter) during this first step. Sub-grouping allows the reader to gain a broad overview of the research that has been identified during the review process. The review team will meet at the start of this process to discuss potential sub-groupings and after all studies have been grouped to discuss how well the sub-groupings fit the data, what new sub-groupings have emerged and data fit, and study sub-grouping overlap. Any changes to the sub-grouping process will be through consensus which may require reviewer 1 to re-visit this first step and re-group studies.

The second step is to develop a summary for each study. This is where key study aspects, findings, author conclusions, limitations, and quality assessment will be described. Quality aspects such as potential biases, confounding variables, participant characteristics, measures, research design, analytic approaches, and key findings will be summarised. Reviewer 2 will audit the same randomly selected 20% of studies for quality assessment coherence, bias, and accuracy. The final step is where reviewer 1 will compare sub-groups in terms of similarities, differences, and scope to enable a synthesis to be produced. The research team will meet at the end of this synthesis process to discuss the conclusions being drawn from the research evidence with any changes being the result of consensus. Stakeholder consultation will also occur at this point with conclusions being shared with key stakeholders for critique and comment.

## Step 6 –Stakeholder consultation

Stakeholder consultation has moved from an optional aspect to a strongly recommended part of the scoping review process [57, 58]. This review will engage in stakeholder consultation

with consultation focusing on two primary aims. Validation of the scoping review findings is the first with the second being to provide alternative insights, meanings, and implications of the findings. In terms of validation, up to three researchers who have conducted suicidality research will be contacted to engage in a peer review of a draft paper that will overview synthesis and review findings along with the review process. Selection will be based on researchers whose work has not included in the review and who have experience conducting scoping or systematic reviews.

Up to six stakeholders representing online and social media organisations (e.g., Facebook, Twitter), suicide prevention/intervention organisations (e.g., International Association for Suicide Prevention, Suicide Prevention Australia), and lived experience representative bodies (e.g., Roses in the Ocean) will be invited to take part in a zoom focus group where review findings will be presented for consideration from organisational, and usage perspectives. There is no guidance as to the number of stakeholders to engage, thus six has been chosen to be feasible and to allow for two representatives from each broad category to be sought. Stakeholders will be encouraged to consider alternative insights, meanings, and implications of the results. Other stakeholders may also be offered the opportunity to provide written feedback and will be selected on the basis of the review findings. For example, if results indicate that most studies have been conducted using online mental health forums, then mental health organisations that manage online forums (e.g., SANE Australia) will be invited to participate.

Reviewer 1 will complete stakeholder selection with the team meeting to ratify selection based on consensus agreement. Additional broader stakeholder consultation will occur through traditional academic outlets with the outcome of the review being submitted for open access publication and for presentation at key suicide prevention or mental health conferences.

## Team composition

Levac et al [57] argue that team composition is critical to the successful conduct and completion of a scoping review. Reviewer 2 has conducted one scoping review as lead author, has published one scoping review protocol, and has experience in suicidality research. Reviewer 3 has expertise in qualitative research and has been involved as a reviewer of a previous scoping review focusing on suicide capability. In addition, reviewer 3 is a registered psychologist with over 25 years as a practitioner. Reviewer 1 has expertise in both qualitative and quantitative suicidality research, has supervised six scoping review studies which includes two published protocols and two published reviews, is currently supervising a further two scoping reviews, and has extensive experience in policy development. Reviewer 1 is also a registered psychologist with over 20 years' experience as a practitioner.

## Discussion

The purpose of the proposed scoping review is to systematically map evidence relating to online STBs talk. In order for the proposed review to make a meaningful contribution to the substantive area, it must be free from bias, rigorous, transparent, and be reproducible. Thus this a priori protocol has been presented for critical assessment and feedback to ensure that it best meets the review's intended purpose.

Twelve databases have been identified as being most pertinent to the study. Two independent reviewers will search these in a methodical and reproducible manner to identify relevant studies. A quality assessment will be undertaken in two phases: first, studies will be limited to peer reviewed studies; second, included studies will be assessed using modified evidence-based quality assessment checklists. By synthesising findings and identifying research gaps and study

limitations, practitioners, mental health organisations and researchers will have an evidence-base that can be utilised to develop practices, make informed decisions, and plan future studies that are more likely to advance our understandings of suicide and how people use the Internet and social media for suicide-related purposes.

The decision to limit studies to peer review does introduce a limitation into the review process. Excluding grey literature from scoping reviews is becoming more common place given the difficulties in searching and identifying appropriate grey literature in a manner that is systematic and reproducible [86]. We do recognise that peer-review in and of itself does not mean that the published study is of high quality [87], hence why a quality assessment has also been included in the proposed review. Another limitation is restricting studies to those published or translated into English. Where there is no translation, studies will be included in a table and will be noted as being a potential limitation to the review's findings.

Identifying and charting the empirical evidence relating to online STBs talk is likely to lead to a better understanding of what is being said, how it is being said, and what is occurring in online spaces when users talk about their STBs. How users engage with, and in these spaces can then be used to inform the development of online spaces that have a suicide prevention and intervention focus. This will encourage a shift from retrospective to prospective approaches to online STB research. That is, from a risk/benefit/content research and understandings of online suicide spaces and social media platforms to prospective, in situ research and knowledge that captures the complexity of suicide as it happens in real time online. In this way the process and experience of suicide can be more fully explored [88].

## Supporting information

**S1 File. Electronic databases and their associated search strings.**
(DOCX)

**S2 File. Microsoft excel review audit trail.**
(XLSX)

**S3 File. Data charting template.**
(XLSX)

**S4 File. Measurement quality assessment.**
(DOCX)

**S5 File. PRISMA-ScR checklist.**
(DOCX)

**S1 Fig. PRISMA flow chart.**
(ZIP)

## Acknowledgments

We would like to thank Sue Griffits, Jens Christensen, and Mark Papinczak for their helpful comments on earlier versions of this work.

## Author Contributions

**Conceptualization:** Andrea Lamont-Mills, Steven A. Christensen.

**Formal analysis:** Andrea Lamont-Mills, Luke T. Bayliss.

**Investigation:** Andrea Lamont-Mills, Luke T. Bayliss.

**Methodology:** Andrea Lamont-Mills, Luke T. Bayliss, Steven A. Christensen.

**Writing – original draft:** Andrea Lamont-Mills.

**Writing – review & editing:** Luke T. Bayliss, Steven A. Christensen.

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
