## [Decision Letter · Decision Letter 0]

12 Jul 2022

PONE-D-22-09022Talking about suicide online: A scoping review protocol.PLOS ONE

Dear Dr. Lamont-Mills,

Thank you for submitting your manuscript to PLOS ONE. After careful consideration, we feel that it has merit but does not fully meet PLOS ONE’s publication criteria as it currently stands. Therefore, we invite you to submit a revised version of the manuscript that addresses the points raised during the review process.

We look forward to receiving your revised manuscript.

Kind regards,

Sarah A. Arias

Academic Editor

PLOS ONE

Journal Requirements:

Reviewers' comments:

Reviewer's Responses to Questions

**Comments to the Author**

1. Does the manuscript provide a valid rationale for the proposed study, with clearly identified and justified research questions?

Reviewer #1: Partly

Reviewer #2: Yes

2. Is the protocol technically sound and planned in a manner that will lead to a meaningful outcome and allow testing the stated hypotheses?

Reviewer #1: Yes

Reviewer #2: Yes

3. Is the methodology feasible and described in sufficient detail to allow the work to be replicable?

Reviewer #1: Yes

Reviewer #2: Yes

4. Have the authors described where all data underlying the findings will be made available when the study is complete?

Reviewer #1: Yes

Reviewer #2: Yes

5. Is the manuscript presented in an intelligible fashion and written in standard English?

Reviewer #1: Yes

Reviewer #2: Yes

6. Review Comments to the Author

You may also provide optional suggestions and comments to authors that they might find helpful in planning their study.

Reviewer #1: This paper describes a protocol for a scoping review focussed on how people talk about suicide thoughts and behaviours online. The authors should be commended for such a detailed protocol which clearly adheres to an open science framework.

General comments: I think the language of this study needs refining. You say that the review is focussed on 'how people talk about STBs online' but in reality you are only measuring via what medium they talk about it (e.g. comments, DMs, with moderators / with others). This is really only one part of 'how' people talk about it - e.g. 'how people talk about it' implies a deeper conversation or language analysis that will explore what meanings people give / the language used to discuss STBs online. I think this distinction needs to be made clearer throughout and the title and other references needs to be more specific e.g. 'via what medium do people talk about STBs online / on social media?'

Introduction:

Line 60-61 - Can you give some examples of 'text' based language (e.g. commenting on content, private messaging, forum / thread discussions)

Lines 75-84 - I think you need to expand on the possible negative consequences of talking online - not just for the user, but for people seeing / responding to the post also. E.g. possible contagion / imitation effects of talking about suicide online. You should also mention the #chatsafe guidelines here which aim to mitigate against these adverse effects

Lines 85-92 - this paragraph feels similar to the paragraph which starts on line 55. Can you either make it clear how they are separate or merge the two paragraphs together.

I think the introduction would benefit from a definition of what you are defining as suicide (e.g. vs self-harm) thoughts / behaviours (I know this is briefly commented upon in the methods but I feel it warrants further explanation in setting the context for the study) as well as what you are classifying as 'internet' (e.g. search engines, social media, information / education websites, forums).

Methods:

I think a comprehensive list is needed to operationalise 'how' people talk about their STBs. You have given examples, but I think to be more rigorous you should have a pre-defined list of criteria that you are looking for in the scoping review.

I recommend taking out of the table the part about opinion pieces etc being excluded as it doesn't fit here very well (but important information to include - I recommend placing below or above the table).

You have specified Twitter, Instagram, Blogs, MySpace and Facebook - does this mean you will exclude other social media sites? E.g. Snapchat, Weibo etc. It might be best as suggested above, to give a definition of what you are terming 'online' spaces and then simply state here that you will include 'social media'.

Line 202 - please state which date searches will be limited from / until

Line 210-211 - it is best practice to include both peer review and grey literature in a review. I appreciate you have justified why not to include grey literature, though my recommendation would still be to include it. Given this is a scoping review, it will still be good to have an idea of what literature is available, even taking into account the limitations you discuss with this not being systematic.

Line 338 - I recommend the authors consider using this more recent checklist for qualitative research: Long, H. A., French, D. P., & Brooks, J. M. (2020). Optimising the value of the critical appraisal skills programme (CASP) tool for quality appraisal in qualitative evidence synthesis. Research Methods in Medicine & Health Sciences, 1(1), 31-42.

Line 339 - can you list the other checklists that are deemed to be of use in this study (e.g. case studies, longitudinal studies etc)

Line 341 - please outline fully the modifications made to the checklists

Line 362 - is there benefit in synthesising the quantitative and qualitative results separately? If not, please justify why they will be synthesised together

Line 369 - please change language from 'ideators' as this is considered stigmatising by some

Line 384 - I would suggest that the sinthesis process (e.g. comparing subgroups) would be more rigorous if conducted as a team, as opposed to only reviewer 1. In fact, involving people here with lived experience would benefit the study greatly.

Line 407 - I strongly recommend that you add a third group of stakeholders - users of internet for suicide thoughts / behvaiours (or even just users of the internet in general)

Discussion:

Minor errors to be corrected:

Abstract, line 24 - change 'synthesises' to 'syntheses'

Intro, line 68 - remove 'like'

In table - in concept line, middle line, add 'data such AS single...'

Methods, line 298 - change 'combing' to 'combining'

Reviewer #2: Thank you for the opportunity to review this paper. It is a very interesting research area and a timely contribution to the field. The authors have presented a well-written and carefully considered protocol for this scoping review, and it is clear that the team are very well placed to conduct this work.

My main feedback for the paper is below and relates to terminology/context – this will be important in guiding the search terms and inclusion criteria for conducting the scoping review:

1. ‘Online suicidal thought and/or behaviour talk’ – this term needs to be more comprehensively defined. ‘Talk’ alone does not take into account the complexity, nuance and online social norms related to communication more generally. Reviewing ‘talk’ alone (without it being more clearly defined, I understand this to be text only) may not represent the way users communicate with one another online with the exclusion of image, video, memes etc. This will be a limitation of the review, and therefore a limitation of the conclusions drawn. Further thinking about the way many use social media or online spaces to communicate, is it possible to broaden the search to include more specific behaviours associated with certain platforms (e.g., Instagram, TikTok, Snapchat). Images in this sense may be a core component of the interaction or help-seeking / support providing behaviour.

2. Adding to the above, further information on why ‘talk’ has been selected would strengthen the rationale of this work – i.e., why is language important (particularly in this field) and why might that be different or the same to the offline context? Defining what is meant by ‘talk’ for the purposes of this research will help contextualise this.

3. The limitation relating to only including peer-reviewed literature has been acknowledged, but perhaps overlooked slightly. Given the novelty of this area and given that so many resources related to suicide-related content online are not peer-reviewed (particularly those created by the platforms themselves, which govern the way users can/cannot use these spaces to communicate about suicide), there may be a lot of very useful and important information excluded.

4. The exclusion criteria relating to self-harm must be further defined. How will you distinguish between the two (See Table 1). What is and is not meant by STBs? How will you distinguish between self-harm, NSSI and suicide for the purposes of this research?

5. Table 1: Concept – Can interactions also include photos/videos/live streams? Or only text?

6. Line 131: The authors touch on the limitation of research to date that focuses on the use of the internet and social media as a tool for suicide prevention. I think this is too simplistic and does not appropriately discuss the work that has been conducted in this area.

More minor feedback:

1. An acknowledgement of the age range of users being included in this review would be helpful – i.e., are all ages being included? Are there developmental differences in the way younger people versus older people use these spaces for suicide talk?

2. The authors touch on the majority of research conceptualising users as ‘passive information seekers’, and it would be good to see more of a discussion around the active nature of social media behaviour. The introduction perhaps overlooks some of the work that has focused on the supportive nature of online spaces/social media, as well as the online behaviours relating to self-presentation etc., that again help contextualise why the online space is important to consider when it comes to STBs.

3. Some typo’s throughout: e.g., line 104/105, 338 (parentheses not closed), 397, 402

4. Cyberpsychology research journal that should be included: Computers in Human Behaviour

Overall, this protocol outlines a scoping review that will identify gaps within this research topic, with the potential for highlighting future research directions within the field of online behaviour and suicide prevention.

7. PLOS authors have the option to publish the peer review history of their article (what does this mean?). If published, this will include your full peer review and any attached files.

Reviewer #1: **Yes: **Dr Laura Hemming

Reviewer #2: No

---

## [Author Response · Author response to Decision Letter 0]

23 Aug 2022

we have provided this as a document as well

Responses to Reviewers

We thank the reviewers for their encouraging and insightful comments and taking the time to review our work. We feel that the comments and considerations have made this a stronger protocol. Please see our responses below. Please note that the line numbers refer to the Revised Manuscript with Track Changes document so that the reviewers can see the changes.

Reviewer 1 comments Author’s responses are underneath each comment

I think the language of this study needs refining. You say that the review is focussed on 'how people talk about STBs online' but in reality you are only measuring via what medium they talk about it (e.g. comments, DMs, with moderators / with others). This is really only one part of 'how' people talk about it - e.g. 'how people talk about it' implies a deeper conversation or language analysis that will explore what meanings people give / the language used to discuss STBs online. I think this distinction needs to be made clearer throughout and the title and other references needs to be more specific e.g. 'via what medium do people talk about STBs online / on social media?'

 We appreciate the opportunity to clarify our thinking on this matter. This scoping review seeks to map research that has examined the ways in which people talk about STBs in online spaces. Ways is used to encompass research that has examined what people say but also the how. Research that focuses on either the what or the how, can capture the meanings that individuals give to their STBs. Our use of how was reflective of its wider use but acknowledge that this could be confusing for readers. In essence we are seeking to map research that has focused on online STBs talk.T o address this, we have refined the title (see line 4 “Online suicidal thoughts and/or behaviours talk: A scoping review”) and research question (see lines 213-215).

“The aim of the proposed scoping review is to systematically map the empirical literature that has investigated STBs online STBs talk.”

As a result, we have revised the introduction and method section to make this point clearer. We have used the term online STBs talk throughout the paper, removed any references to how people talk about STBs, and re-arranged the introduction so what is meant by online talk appears earlier. This allows for clarity about the review’s focus (see lines 83-89)

“Moving to online talk, we define online talk as the digital text-based language people use when communicating to and with others, either in real-time or asynchronously [14]. This is via mediums that include but are not limited to online forums, direct messaging apps or blogs. Online talk therefore includes the posts and reply comments people make on such forums, the messages that they send on direct messaging apps, and what they write on blogs and in blog comments. In this sense online STBs talk encompasses written text about STBs be this one’s own or others.”

Introduction:

Line 60-61 - Can you give some examples of 'text' based language (e.g. commenting on content, private messaging, forum / thread discussions). This has been included in lines 83-89

“Moving to online talk, we define online talk as the digital text-based language people use when communicating to and with others, either in real-time or asynchronously [14]. This is via mediums that include but are not limited to online forums, direct messaging apps or blogs. Online talk therefore includes the posts and reply comments people make on such forums, the messages that they send on direct messaging apps, and what they write on blogs and in blog comments. In this sense online STBs talk encompasses written text about STBs be this one’s own or others.”

Lines 75-84 - I think you need to expand on the possible negative consequences of talking online - not just for the user, but for people seeing / responding to the post also. E.g. possible contagion / imitation effects of talking about suicide online. You should also mention the #chatsafe guidelines here which aim to mitigate against these adverse effects This is included in lines 148-150

“Further, there is the potential that reading and/or commenting on others online STBs talk can be triggering as there can be contagion effects of viewing and engaging with others online and with online STBs content [41].”

Lines 85-92 - this paragraph feels similar to the paragraph which starts on line 55. Can you either make it clear how they are separate or merge the two paragraphs together.

 We thank the reviewer for this suggestion. What we have done is removed the reference to connectedness and belonging from the line 55 paragraph (now starting on line 97) and included additional information about comparing online to offline talk as recommended by Reviewer 2. The second referenced paragraph now starts on line 107.

I think the introduction would benefit from a definition of what you are defining as suicide (e.g. vs self-harm) thoughts / behaviours (I know this is briefly commented upon in the methods but I feel it warrants further explanation in setting the context for the study) as well as what you are classifying as 'internet' (e.g. search engines, social media, information / education websites, forums). We have included definitions for the internet and social media as the second paragraph – see lines 48-53. Aspects of this definition has also been included Table 1). 

“There is conjecture about what is meant by the terms internet and social media. Broadly, internet relates to internet-based applications (e.g., chat rooms, online forums, websites) that allow for content creation and sharing with the potentiality of user engagement with the posted content and by default other users [7]. Social media refers to mobile applications (e.g., FaceBook, Twitter) that also enables users to create content and share this with others who can engage with this user generated content and the user themselves [7].”

Definitions relating to STBs occurs on lines 71-83. Aspects of these definitions have also been included in Table 1 for consistency.

“Suicide and STBs are contested terms with little previous consensus on what they encompass and mean [13]. Drawing upon the recent work of De Leo and colleagues [13], suicide is a fatal act that is carried out with knowledge of this fatality. Suicidal ideation encompasses thoughts of killing oneself where there may or may not be an intention to take one’s own life. It is the absence of behaviour that distinguishes suicidal thoughts from suicide and suicidal behaviour, noting that suicide in and of itself is a behaviour. Taking a broad perspective on suicidal behaviour, this refers to having made plans or preparing to take one’s own life that includes the how and when of this but also extends to having made a suicide attempt [13]. It is perhaps with self-harm that most disagreement arises, centring around the issue of intention. For our purposes, self-harm is when an individual engages in activities that are harmful to themself but there is an absence of an intention to die, thus the activities are non-fatal [13].”

I think a comprehensive list is needed to operationalise 'how' people talk about their STBs. You have given examples, but I think to be more rigorous you should have a pre-defined list of criteria that you are looking for in the scoping review.

 We thank the reviewer for this consideration. Because we have clarified the intent of the scoping review the how is no longer relevant and has been removed. Whilst we note the point about rigour, scoping reviews are iterative and thus inclusion and exclusion criteria can and should evolve over time. A comprehensive pre-defined list of criteria is associated with systematic literature reviews not scoping reviews. We have included a pre-defined list that we believe is of sufficient scope to allow for the first scoping review search iteration to occur.

What the reviewer may have overlooked is the criteria is a result relating to online STBs talk. What this result may look like, and reference, will emerge from the actual review and thus cannot be definitively pre-defined. The key is that we will include any study that has such a result. It does not matter if this is not the focus of the study, it is about results of the study.

I recommend taking out of the table the part about opinion pieces etc being excluded as it doesn't fit here very well (but important information to include - I recommend placing below or above the table). We thank the reviewer for their suggestion. As these are exclusion criteria, they are required to be placed in the Table underneath the exclusion column.

You have specified Twitter, Instagram, Blogs, MySpace and Facebook - does this mean you will exclude other social media sites? E.g. Snapchat, Weibo etc. It might be best as suggested above, to give a definition of what you are terming 'online' spaces and then simply state here that you will include 'social media'. A definition as noted above has been included in both the introduction and method sections to address this point. We have edited the inclusion criteria in Table 1 to make it clearer that the list is not exhaustive. 

“Other spaces are likely to be identified and added to this as an outcome of step 2.”

We have also included Snapchat, Weibo as possible examples in Table 1 and included these in the search strings. Again, a comprehensive pre-defined list is reflective of a systematic review not the iterative nature of a scoping review and we believe at this point the pre-defined list is of sufficient scope to allow for the first scoping review search iteration to occur.

Line 202 - please state which date searches will be limited from / until A date range has been included on line 277 “A date limitation of 1989 onward…”

Line 210-211 - it is best practice to include both peer review and grey literature in a review. I appreciate you have justified why not to include grey literature, though my recommendation would still be to include it. Given this is a scoping review, it will still be good to have an idea of what literature is available, even taking into account the limitations you discuss with this not being systematic. We thank the reviewer for their comment. We stand by our original decision and have justified this further in the method section. Issues go beyond being systematic and reproducibility and we have made this clearer on lines 347-359.

“Whilst including a search of the grey literature has been a notable feature of scoping reviews [61, 63], there are no clear guidelines as to what grey literature must be included in a scoping review. Further, there are no best practice recommendations as to how to search for and thus identify all relevant grey literature sources [82]. Currently it is recommended that overlapping search strategies be adopted in order to identify all relevant pieces of grey literature, however this brings with it the probability of also identifying irrelevant literature [82]. Thus, it is not only replicability that becomes an issue but as Peters and colleagues [63] posit the scope and breadth of the review needs to be balanced alongside the feasibility of completing the review with the resources available. We recognise that the exclusion of grey literature will impose limitations on review findings, and this will be noted in limitation discussions. As per Peters et al.[63, 83] recommendation when deciding to not include particular searches, we have justified, detailed, and made transparent our decision to not include grey literature and recognise this as a limitation.”

Line 338 - I recommend the authors consider using this more recent checklist for qualitative research: Long, H. A., French, D. P., & Brooks, J. M. (2020). Optimising the value of the critical appraisal skills programme (CASP) tool for quality appraisal in qualitative evidence synthesis. Research Methods in Medicine & Health Sciences, 1(1), 31-42. We thank the review for their suggestion. The JBI checklist that we are now using is the 2020 version. We have checked for changes and the changes do not impact the adaptions we have already made. These can be carried over to the 2020 JBI checklist. Moreover, when we reviewed the recommended checklist, it covers the same content as the JBI checklist thus there is no additional benefit in using the Long et al checklist. Further, using different study design checklists that have been developed using the same methodology strengthens the rigor of the extraction and synthesis process.

Line 339 - can you list the other checklists that are deemed to be of use in this study (e.g. case studies, longitudinal studies etc).

 We have included case reports (see lines 429-430 “Checklist Analytical Cross-Sectional Studies [86], case reports the JBI Appraisal Checklist Analytical Case Report[87])”). Other study design checklists will be reflective of search results. From our trial searches cross-sectional, case studies, and qualitative research designs are most frequent. Thus we feel that at this point this is sufficient for the protocol.

Line 341 - please outline fully the modifications made to the checklists

 This is now on lines 434-441. We have also included the measurement assessment as a supplementary file. “These changes are based on Macrynikola et al [52] recommendations for selection, measurement and confounds. In terms of selection: did the study include clear sample descriptions/definitions such as age, gender orientation, nationality, sexual orientation, geographical location; was a participation rate included (if relevant) and was the recruitment criteria applied appropriately and uniformly; and was a justification of the sample size included. M, measurement focused on the quality of measurement used in each study with a rating of high to low given (see Supplementary File 6) with confounds focusing on whether basic and potential confounds that have been identified in the literature were taken into consideration and discussed.”

Line 362 - is there benefit in synthesising the quantitative and qualitative results separately? If not, please justify why they will be synthesised together Please forgive us but we are not quite clear what the reviewer is trying to say as we have not seen scoping reviews that synthesis qualitative or quantitative results separately.

We have now included the benefits of using such an approach as seen on lines 464-466.

“…evidence as it allows for a standardise reporting of each type of study [88]. This is turn makes it easier for differences and similarities between study contexts and characteristics to be identified [89].”

Line 369 - please change language from 'ideators' as this is considered stigmatising by some We note the reviewers concerns however this is the language used in studies that will form the review. The sub-groups are and will be reflective of participant groups included in the identified studies. It is also currently used in research as a descriptor. We also note that common language guidelines do not suggest avoiding this term. Thus, at this point in time we will use this term making clear that the term is a study related term.

Line 384 - I would suggest that the sinthesis process (e.g. comparing subgroups) would be more rigorous if conducted as a team, as opposed to only reviewer 1. In fact, involving people here with lived experience would benefit the study greatly. We thank the reviewer for the suggestion of considering another reviewer. This is a scoping review not a systematic review. In systematic reviews two or more reviewers are required for the whole process, in a scoping review that adopts the Arskey and O’Malley approach only one reviewer is required for extraction and synthesis.

To address the concerns raised we had already included a random audit of extracted information (see line 416). Further as stated in the method section, the team will meet and discuss the conclusions (see lines 482-489). These have been included to ensure that any bias introduced into the extraction and synthesis process is identified.

We note the suggestion of including someone with lived experience in the study. If this were an actual study examining how someone talks about their STBs online then including someone with lived experience of talking about their STBs online may be appropriate. However, this is a scoping review, it is a systematic review of research. Findings and conclusions are based on what research has already been conducted. Conclusions and future research is based on identified research gaps. All that is recommended must be based on the systematic review of evidence. We cannot see what additional perspective someone with lived experience would bring given the lived experience perspective can and is being introduced in the key stakeholder section on Reviewer 1’s recommendation. 

With that we can see the benefit of including a stakeholder user group where lived experience perspectives can be used to critically examine the claims made in the review.

Line 407 - I strongly recommend that you add a third group of stakeholders - users of internet for suicide thoughts / behvaiours (or even just users of the internet in general) We thank the reviewer for this suggestion. This prompted us to review all stakeholders for alignment with our aim of validating our findings and providing alternative insights, meanings, and implications of the findings.

As such we have removed health practitioners and key government bodies as they are typically included when reviewing more practice or policy related topics. We have also included examples of key stakeholders and included lived experience representative bodies to reflect the suggestion. 

Our decision to go with lived experience representative bodies reflects a pragmatic approach to identifying users. We will be clear that we are wanting people with experience of using the internet and social media for suicide-related reasons when we approach these bodies for someone who can speak on behalf of others.

See lines 500-521 for these changes.

“Up to six stakeholders representing online and social media organisations (e.g., FaceBook, Twitter), suicide prevention/intervention organisations (e.g., International Association for Suicide Prevention, Suicide Prevention Australia), and lived experience representative bodies (e.g., Roses in the Ocean) will be invited to take part in a zoom focus group where review findings will be presented for consideration from organisational, and usage perspectives. There is no guidance as to the number of stakeholders to engage, thus six has been chosen to be feasible and to allow for two representatives from each broad category to be sought. Stakeholders will be encouraged to consider alternative insights, meanings, and implications of the results. Other stakeholders may also be offered the opportunity to provide written feedback and will be selected on the basis of the review findings. For example, if results indicate that most studies have been conducted using online mental health forums, then mental health organisations that manage online forums (e.g., SANE Australia) will be invited to participate. ”

Minor errors to be corrected:

Abstract, line 24 - change 'synthesises' to 'syntheses'

Intro, line 68 - remove 'like' 

Completed

Completed

In table - in concept line, middle line, add 'data such AS single...' We have re-phrased this sentence in relation to a previous comment. This now reads “Data that would have produced this result would include but are not limited to…”

Methods, line 298 - change 'combing' to 'combining' Completed

Reviewer 2 comments 

 ‘Online suicidal thought and/or behaviour talk’ – this term needs to be more comprehensively defined. ‘Talk’ alone does not take into account the complexity, nuance and online social norms related to communication more generally. Reviewing ‘talk’ alone (without it being more clearly defined, I understand this to be text only) may not represent the way users communicate with one another online with the exclusion of image, video, memes etc. This will be a limitation of the review, and therefore a limitation of the conclusions drawn. Further thinking about the way many use social media or online spaces to communicate, is it possible to broaden the search to include more specific behaviours associated with certain platforms (e.g., Instagram, TikTok, Snapchat). Images in this sense may be a core component of the interaction or help-seeking / support providing behaviour. We thank the reviewer for this comment. We have moved the definition of online talk earlier in the introduction so that the reader is clearer as to what we mean by this term (see lines 83-89). We have also defined what we mean by suicide, suicidal thoughts and behaviours earlier as well so that the reader understands the position we are taking up (see lines 71-82).

“Suicide and STBs are contested terms with little previous consensus on what they encompass and mean [13]. Drawing upon the recent work of De Leo and colleagues [13], suicide is a fatal act that is carried out with knowledge of this fatality. Suicidal ideation encompasses thoughts of killing oneself where there may or may not be an intention to take one’s own life. It is the absence of behaviour that distinguishes suicidal thoughts from suicide and suicidal behaviour, noting that suicide in and of itself is a behaviour. Taking a broad perspective on suicidal behaviour, this refers to having made plans or preparing to take one’s own life that includes the how and when of this but also extends to having made a suicide attempt [13]. It is perhaps with self-harm that most disagreement arises, centring around the issue of intention. For our purposes, self-harm is when an individual engages in activities that are harmful to themself but there is an absence of an intention to die, thus the activities are non-fatal [13].”

“Moving to online talk, we define online talk as the digital text-based language people use when communicating to and with others, either in real-time or asynchronously [14]. This is via mediums that include but are not limited to online forums, direct messaging apps or blogs. Online talk therefore includes the posts and reply comments people make on such forums, the messages that they send on direct messaging apps, and what they write on blogs and in blog comments. In this sense online STBs talk encompasses written text about STBs be this one’s own or others.”

We note the point about communication, and we do not dispute this. We argue that text-based and visual communication are two different styles of communication that warrant separate consideration. We have made this point on lines 89-96 and why we are, at this point, focusing on text communication only.

“Taking up an online STBs talk focus does not mean that people cannot and do not communicate STBs visually (e.g., via images, videos, memes [15, 16]). Communicating via text and visual methods draws upon different understandings and competencies and are often used for different communication purposes [17]. They are two related but distinctive ways of communicating and as such visual communication of STBs requires its own separate consideration. This is particularly so as many online spaces restrict STBs images, videos, and memes being shared [18] but are less restrictive regarding online STBs talk.”

We recognised that this will be a limitation of the review, but it is a limitation we are prepared to accept. In fact, we see visual communication as a separate scoping review given the embodied nature of communication.

We feel that we were not clear in Table 1 that we are not excluding studies where participants use an image and talk about STBs in reference to the image in an online space. What we are excluding is studies that only analyse the visual image rather than any textual commentary on the image. Additional clarity has been included in Table 1 to reinforce this point. 

2. Adding to the above, further information on why ‘talk’ has been selected would strengthen the rationale of this work – i.e., why is language important (particularly in this field) and why might that be different or the same to the offline context? Defining what is meant by ‘talk’ for the purposes of this research will help contextualise this.

 We have included reference to the importance of talk through reference to various safe language guidelines and the number of suicide prevention training programs focusing on how to talk with someone who is in a heightened state of suicidal desire (see lines 125-133).

“Moreover, the importance of talk is reflected in safe language guidelines. These have been developed to ensure that the stigma and shame surrounding suicide is not being perpetuated through language (see #chatsafe guidelines [32]; Mindframe language guidelines [33]; CAMH words matter: Suicide language guide [34]). Further, asking and talking about suicide is noted as being difficult with this being reflected in the number of suicide prevention training programs specifically designed to assist ‘ordinary’ people to talk with someone who is in a state of heighted suicidal desire (e.g., safetalk [35], calm conversations [36] and zero suicide alliance suicide awareness training [37]).”

We have also included how talking online and offline about STBs may be different – see lines 101-105.

“Talking about STBs online is, therefore, likely to be different to how STBs are talked about in offline spaces. Individuals who are in a heightened state of suicidal desire often withhold or are reluctant to share information in face-to-face or real-time settings [21] and feel they are able to discuss matters that concern them more freely in online spaces [20, 22].”

3. The limitation relating to only including peer-reviewed literature has been acknowledged, but perhaps overlooked slightly. Given the novelty of this area and given that so many resources related to suicide-related content online are not peer-reviewed (particularly those created by the platforms themselves, which govern the way users can/cannot use these spaces to communicate about suicide), there may be a lot of very useful and important information excluded. We thank the reviewer for this comment. This is a scoping review of research that is focusing on online STBs talk. We are not looking to identify resources that govern the way users engage in online spaces as these are resources, not research studies, and thus they are not relevant to the proposed scoping review. A focus on resources would be a different scoping review with different research questions and objectives.

4. The exclusion criteria relating to self-harm must be further defined. How will you distinguish between the two (See Table 1). What is and is not meant by STBs? How will you distinguish between self-harm, NSSI and suicide for the purposes of this research? We have now included a definition of STBs and also self-harm (see lines 71-82).

“Suicide and STBs are contested terms with little previous consensus on what they encompass and mean [13]. Drawing upon the recent work of De Leo and colleagues [13], suicide is a fatal act that is carried out with knowledge of this fatality. Suicidal ideation encompasses thoughts of killing oneself where there may or may not be an intention to take one’s own life. It is the absence of behaviour that distinguishes suicidal thoughts from suicide and suicidal behaviour, noting that suicide in and of itself is a behaviour. Taking a broad perspective on suicidal behaviour, this refers to having made plans or preparing to take one’s own life that includes the how and when of this but also extends to having made a suicide attempt [13]. It is perhaps with self-harm that most disagreement arises, centring around the issue of intention. For our purposes, self-harm is when an individual engages in activities that are harmful to themself but there is an absence of an intention to die, thus the activities are non-fatal [13].”

In Table 1 we have included measures of self-harm that would enable the team to distinguish between self-harm, NSSI and STBs.

“Thus, studies that have measures of self-harm that have items relating to STBs (e.g., Self-Harm Inventory [69]) would be included but studies that use measures that do not (e.g, Self-Harm Behavior survey [70]) would not be included. It is the absence of a measure of STB that would exclude a study.”

5. Table 1: Concept – Can interactions also include photos/videos/live streams? Or only text?

 Interactions are textual in nature as per our definition of online STBs talk. However, this does not preclude textual interactions that are about shared images. As noted in our response to the previous point about images, if the analysis of a study is purely a visual analysis, then this study will not be included in this scoping review.

6. Line 131: The authors touch on the limitation of research to date that focuses on the use of the internet and social media as a tool for suicide prevention. I think this is too simplistic and does not appropriately discuss the work that has been conducted in this area. Please forgive us as we are not quite clear what the reviewer is trying to tell us. In this section (now lines 201-210) we are not talking about the limitations of research that has been conducted in the area. Our point is that there are no syntheses (i.e., systematic or scoping reviews of research) that have specifically focused on online STBs talk. That is, the limitation is not that there isn’t research that has been conducted rather the limitation is about the lack of a scoping or systematic review of this research. We have removed the first sentence of that paragraph and also changed text on lines 201 to 209 to make it clearer we are talking about systematic and scoping reviews rather than conducted research studies. 

“Currently there are no syntheses of research studies focusing on such online STBs talk. A lack of a comprehensive overview of evidence means that the potential of the internet and social media platforms to inform suicide prevention and intervention approaches remains under-realised. Further, internet and social media technology has developed and changed rapidly in the past 5 to 10 years with more mental health organisations offering online suicide support. This with the more recent development of peer-led online suicide prevention programs [60], raises questions about the utility of means that prior systematic and scoping reviews as platforms for understanding how to best engage people who are experiencing STBs and turning to online places for support and help.”

More minor feedback:

1. An acknowledgement of the age range of users being included in this review would be helpful – i.e., are all ages being included? Are there developmental differences in the way younger people versus older people use these spaces for suicide talk?

Such a statement was in Table 1 under the Mnemonic parameters column. However, we note that this was not well phrased so this now reads: 

“Individuals of all ages, genders, sexual orientations, or nationalities...”

Whether there are developmental differences will be a finding of the review and thus we cannot make any statement about this at this point in time.

2. The authors touch on the majority of research conceptualising users as ‘passive information seekers’, and it would be good to see more of a discussion around the active nature of social media behaviour. The introduction perhaps overlooks some of the work that has focused on the supportive nature of online spaces/social media, as well as the online behaviours relating to self-presentation etc., that again help contextualise why the online space is important to consider when it comes to STBs. We have re-phrased this section of the introduction to make our point clearer. We are not overlooking the supportive work rather our point here is that this research focuses on the individual user rather than on their engagement in and with the space. See lines 54-70 for this added clarity and softening of language.

“Whilst the above is suggestive of the interactive nature of both online spaces, the internet and social media are often portrayed as being primarily information sources or repositories [8]. That is, they are online spaces where users go for information and support. Here users are conceptualised as passive information/support seekers [9], which is sometimes reinforced by online suicidology research. This is because individual user experiences or perceptions are privileged, with research typically focusing on the impact that using the internet and social media has on the individual user’s suicidal thoughts and/or behaviours (STBs). Yet these online spaces are more than simply information sources or repositories, they are where users go to engage and interact (i.e., talk) with others and to share their experiences of STBs [8]. In suicidology research the engagement and interaction of users in these online spaces is often overlooked with preference being given to user impact of going online for suicide-related reasons. This is not dismissing the research that has focused on the supportive nature of online spaces for those experiencing STBs (see [10-12]). Rather the point is such research has not necessarily been concerned with actual online STBs talk or how users engage and interact with others in these online spaces. Instead, most of this research focuses on individual user benefits or negative impacts”

3. Some typo’s throughout: e.g., line 104/105, 338 (parentheses not closed), 397, 402

 Thank youfor pointing these out, we do appreciate this. We have checked and corrected typo’s throughout the document as noted on the revised track changed version.

4. Cyberpsychology research journal that should be included: Computers in Human Behaviour This has now been added at line 330.

Journal formatting 

The manuscript was developed using the template.

Upon re-submitting your revised manuscript, please upload your study’s minimal underlying data set as either Supporting Information files or to a stable, public repository and include the relevant URLs, DOIs, or accession numbers within your revised cover letter. For a list of acceptable repositories, please see http://journals.plos.org/plosone/s/data-availability#loc-recommended-repositories. Any potentially identifying patient information must be fully anonymized. As there are no results in a scoping review protocol this statement does not apply for this manuscript.

Done

---

## [Decision Letter · Decision Letter 1]

14 Oct 2022

Online suicidal thoughts and/or behaviours talk: A scoping review.

PONE-D-22-09022R1

Dear Dr. Lamont-Mills,

We’re pleased to inform you that your manuscript has been judged scientifically suitable for publication and will be formally accepted for publication once it meets all outstanding technical requirements.

Kind regards,

Sarah A. Arias

Academic Editor

PLOS ONE

Reviewers' comments:

Reviewer's Responses to Questions

**Comments to the Author**

1. Does the manuscript provide a valid rationale for the proposed study, with clearly identified and justified research questions?

Reviewer #2: Yes

2. Is the protocol technically sound and planned in a manner that will lead to a meaningful outcome and allow testing the stated hypotheses?

Reviewer #2: Yes

3. Is the methodology feasible and described in sufficient detail to allow the work to be replicable?

Reviewer #2: Yes

4. Have the authors described where all data underlying the findings will be made available when the study is complete?

Reviewer #2: Yes

5. Is the manuscript presented in an intelligible fashion and written in standard English?

Reviewer #2: Yes

6. Review Comments to the Author

You may also provide optional suggestions and comments to authors that they might find helpful in planning their study.

Reviewer #2: The authors have responded appropriately to all of the comments raised by the reviewers. They have carefully considered the feedback provided and made necessary edits to their paper.

7. PLOS authors have the option to publish the peer review history of their article (what does this mean?). If published, this will include your full peer review and any attached files.

Reviewer #2: No

---

## [Editor Report · Acceptance letter]

18 Oct 2022

PONE-D-22-09022R1 

Online suicidal thoughts and/or behaviours talk: A scoping review protocol. 

Dear Dr. Lamont-Mills:

I'm pleased to inform you that your manuscript has been deemed suitable for publication in PLOS ONE. Congratulations! Your manuscript is now with our production department. 

Kind regards, 

on behalf of

Dr. Sarah A. Arias 

Academic Editor

PLOS ONE